# MINIGPT-4:
# ENHANCING VISION-LANGUAGE UNDERSTANDING WITH ADVANCED LARGE LANGUAGE MODELS

**Deyao Zhu**\*, **Jun Chen**\*, **Xiaoqian Shen, Xiang Li, Mohamed Elhoseiny**
King Abdullah University of Science and Technology
`{deyao.zhu,jun.chen,xiaoqian.shen,`
`xiang.li.1,mohamed.elhoseiny}@kaust.edu.sa`

## ABSTRACT

The recent GPT-4 has demonstrated extraordinary multi-modal abilities, such as directly generating websites from handwritten text and identifying humorous elements within images. These features are rarely observed in previous vision-language models. However, the technical details behind GPT-4 continue to remain undisclosed. We believe that the enhanced multi-modal generation capabilities of GPT-4 stem from the utilization of sophisticated large language models (LLM). To examine this phenomenon, we present MiniGPT-4, which aligns a frozen visual encoder with a frozen advanced LLM, Vicuna, using one projection layer. Our work, for the first time, uncovers that properly aligning the visual features with an advanced large language model can possess numerous advanced multi-modal abilities demonstrated by GPT-4, such as detailed image description generation and website creation from hand-drawn drafts. Furthermore, we also observe other emerging capabilities in MiniGPT-4, including writing stories and poems inspired by given images, teaching users how to cook based on food photos, and so on. In our experiment, we found that the model trained on short image caption pairs could produce unnatural language outputs (e.g., repetition and fragmentation). To address this problem, we curate a detailed image description dataset in the second stage to finetune the model, which consequently improves the model's generation reliability and overall usability. Our code, pre-trained model, and collected dataset are available at https://minigpt-4.github.io/.

## 1 INTRODUCTION

In recent years, large language models (LLMs) have experienced rapid advancements (Ouyang et al., 2022; OpenAI, 2022; Brown et al., 2020; Scao et al., 2022a; Touvron et al., 2023; Chowdhery et al., 2022; Hoffmann et al., 2022). With exceptional language understanding capabilities, these models can perform a variety of intricate linguistic tasks in a zero-shot manner. Notably, GPT-4, a large-scale multimodal model, has been recently introduced and demonstrated several impressive capabilities of vision-language understanding and generation (OpenAI, 2023). For example, GPT-4 can produce detailed and accurate image descriptions, explain unusual visual phenomena, and even construct websites based on handwritten text instructions.

Although GPT-4 has exhibited remarkable vision language capabilities, the methods behind its exceptional abilities are still a mystery (OpenAI, 2023). We believe that these impressive skills may stem from the utilization of a more advanced large language model (LLM). LLMs have demonstrated various emergent abilities, as evidenced in GPT-3's few-shot prompting setup (Brown et al., 2020) and the findings of Wei *et al.* (2022) (Wei et al., 2022). Such emergent properties are hard to find in smaller-scale models. It is conjectured that these emergent abilities are also applicable to multi-modal models, which could be the foundation of GPT-4's impressive visual description capabilities.

To substantiate our hypothesis, we present a novel vision-language model named MiniGPT-4. It utilizes an advanced large language model (LLM), Vicuna (Chiang et al., 2023), which is built upon

---

\*equal contribution

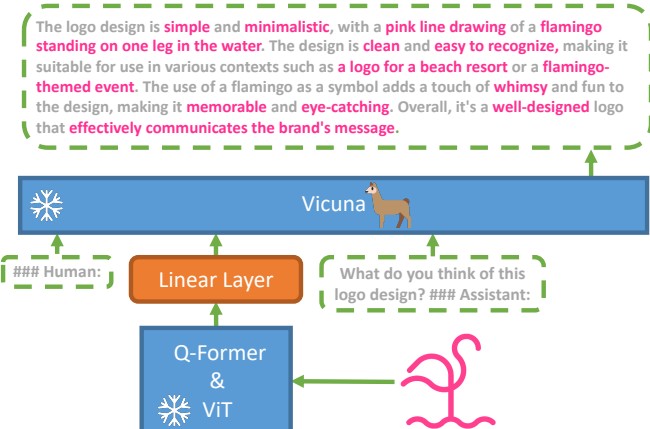

Figure 1: **The architecture of MiniGPT-4.** It consists of a vision encoder with a pretrained ViT and Q-Former, a single linear projection layer, and an advanced Vicuna large language model. MiniGPT-4 only requires training the linear projection layer to align the visual features with the Vicuna.

LLaMA (Touvron et al., 2023) and reported to achieve 90% of ChatGPT's quality as per GPT-4's evaluation, as the language decoder. In terms of visual perception, we employ the same pretrained vision components of BLIP-2 (Li et al., 2023c) that consists of a ViT-G/14 from EVA-CLIP (Fang et al., 2022) and a Q-Former network. MiniGPT-4 adds a single projection layer to align the encoded visual features with the Vicuna language model and freezes all the other vision and language components. MiniGPT-4 is initially trained for 20k steps using a batch size of 256 on 4 A100 GPUs, leveraging a combined image captioning dataset that includes images from LAION (Schuhmann et al., 2021), Conceptual Captions (Changpinyo et al., 2021; Sharma et al., 2018), and SBU (Ordonez et al., 2011) to align visual features with the Vicuna language model. Nevertheless, merely aligning visual features with the language model (LLM) is inadequate to ensure robust visual conversation capabilities, resembling that of a chatbot. The presence of underlying noise in raw image-text pairs can lead to subpar language outputs. Therefore, we collect another 3,500 detailed image description pairs to further fine-tune the model with a designed conversational template in order to improve the naturalness of the generated language and its usability.

In our experiments, we discovered that MiniGPT-4 possesses numerous capabilities similar to those demonstrated by GPT-4. For instance, MiniGPT-4 can generate intricate image descriptions, create websites based on handwritten text instructions, and explain unusual visual phenomena. Furthermore, our findings revealed that MiniGPT-4 also has a variety of other intriguing abilities not showcased in the GPT-4 demonstrations. For example, MiniGPT-4 can directly generate detailed cooking recipes from food photos, write stories or poems inspired by images, write advertisements for products in images, identify problems shown in photos and provide corresponding solutions, and retrieve rich facts about people, movies, or art directly from images, among other capabilities. These abilities are absent in previous vision-language models like Kosmos-1 (Huang et al., 2023) and BLIP-2 (Li et al., 2023c) that use less powerful language models. This further validates that integrating visual features with an advanced language model is one of the keys to enhancing vision-language models. We present a summary of our key findings:

- Our research reveals with compelling evidence that by aligning visual features with advanced large language models like Vicuna, MiniGPT-4 can achieve advanced vision-language capabilities comparable to those exhibited in the GPT-4 demonstrations.

- Our findings suggest that training merely one projection layer can effectively align a pretrained vision encoder with the large language model. Our MiniGPT-4 only requires training approximately 10 hours on 4 A100 GPUs.

- We discovered that simply aligning visual features with large language models using short image caption pairs is not sufficient for developing a well-performing model and leads to unnatural language generation. Further finetuning with a small but detailed image description pairs can address this limitation and significantly improves its usability.

## 2 RELATED WORKS

**Large language models** have experienced tremendous success in recent years due to the scaling up of training data and an increase in the number of parameters. Early models, such as BERT (Devlin et al., 2018), GPT-2 (Radford et al., 2019), and T5 (Raffel et al., 2020), laid the foundation for this progress. Subsequently, GPT-3 (Brown et al., 2020), with a massive scale of 175 billion parameters, was introduced, demonstrating significant breakthroughs across numerous language benchmarks. This development inspired the creation of various other large language models, including Megatron-Turing NLG (Smith et al., 2022), Chinchilla (Hoffmann et al., 2022), PaLM (Chowdhery et al., 2022), OPT (Zhang et al., 2022), BLOOM (Scao et al., 2022b), and LLaMA (Touvron et al., 2023), among others. Wei *et al.* (Wei et al., 2022) further discovered several *emergent abilities*, which appear exclusively in large models. The emergence of these abilities underscores the importance of scaling up in the development of large language models. Moreover, by aligning the pre-trained large language model GPT-3 with human intent, instructions and human feedback, InstructGPT (Ouyang et al., 2022) and ChatGPT (OpenAI, 2022) enable conversational interactions with humans and can answer a wide range of diverse and complex questions. More recently, several open-sourced models, such as Alpaca (Taori et al., 2023) and Vicuna (Chiang et al., 2023), have been developed based on LLaMA (Touvron et al., 2023) and also exhibit similar performance.

**Leveraging Pre-trained LLMs in Vision-Language Tasks.** The use of autoregressive language models as decoders in vision-language tasks has become increasingly popular (Chen et al., 2022; Huang et al., 2023; Yang et al., 2022; Tiong et al., 2022; Alayrac et al., 2022; Li et al., 2023c; 2022; Driess et al., 2023), facilitating cross-modal knowledge transfer. Notable examples include VisualGPT (Chen et al., 2022) and Frozen (Tsimpoukelli et al., 2021), which integrate pre-trained language models for decoding. Flamingo (Alayrac et al., 2022) aligns a vision encoder and language model, excelling in few-shot learning. BLIP-2 (Li et al., 2023c) combines a Flan-T5 (Chung et al., 2022) with Q-Former for efficient alignment. PaLM-E (Driess et al., 2023), with its 562 billion parameters, merges real-world sensor data into an LLM, linking perceptions and languages. GPT-4 (OpenAI, 2023) further advances visual understanding and reasoning after extensive image-text data pre-training. Contemporary works such as LLaVa (Liu et al., 2023a), InstructBLIP (Dai et al., 2023), mPLUG-Owl (Ye et al., 2023), Multimodal-GPT (Gong et al., 2023), and Otter (Li et al., 2023b) align language models with visual encoders using multimodal instruction following datasets. Compared to these methods, MiniGPT-4 demonstrates both data efficiency and parameter efficiency, where only a single linear layer is learnable and the training time is just 10 hours with 4 A100 GPUs. In addition, LLaVa (Liu et al., 2023a), MIMIC-IT (Li et al., 2023a), and M3IT (Li et al., 2023e) collect visual instruction datasets by either generating from ChatGPT or from the human annotators. Such methods require access to image datasets with ground truth image information in text format. Compared to these methods, the visual instruction dataset used in MiniGPT-4 is generated by MiniGPT-4 itself, making data collection model-informed.

LLMs like ChatGPT can enhance vision-language tasks by collaborating with specialized models. Visual ChatGPT (Wu et al., 2023) and MM-REACT (Yang* et al., 2023) show ChatGPT integrating various visual models for complex challenges. ChatCaptioner (Zhu et al., 2023) uses ChatGPT to generate questions for BLIP-2, summarizing image content through dialogue. Video ChatCaptioner (Chen et al., 2023) extends this to video understanding. ViperGPT (Surís et al., 2023) combines an LLM with vision models for visual queries. MiniGPT-4 aligns visual information with the language model directly, avoiding external models.

## 3 METHOD

MiniGPT-4 aims to align visual information from a pretrained vision encoder with an advanced large language model (LLM). Specifically, we utilize the Vicuna (Chiang et al., 2023) as our language decoder, which is constructed upon LLaMA (Touvron et al., 2023) and can perform a wide range of complex linguistic tasks. For visual perception, we employ the same visual encoder as used in BLIP-2 (Li et al., 2023c), a ViT backbone (Fang et al., 2022) coupled with their pre-trained Q-Former. Both language and vision models are open-sourced. We target to bridge the gap between the visual encoder and LLM using a linear projection layer, with an overview of our model displayed in Fig.1.

We use a two-stage training method. First, we pretrain it on a vast set of image-text pairs to learn vision-language skills. Then, we finetune the model using a smaller, high-quality image-text dataset and a conversational template, improving generation reliability and usability.

## 3.1 FIRST PRETRAINING STAGE

In the initial pretraining stage, our model uses a large collection of aligned image-text pairs to gain vision-language knowledge. The output from the projection layer serves as a soft prompt for the LLM, leading it to generate corresponding ground-truth texts. Throughout pretraining, the pretrained vision encoder and LLM remain frozen, with only the linear projection layer undergoing training. We utilize datasets from Conceptual Caption (Changpinyo et al., 2021; Sharma et al., 2018), SBU (Ordonez et al., 2011), and LAION (Schuhmann et al., 2021) for this process. The model undergoes 20,000 training steps with a batch size of 256, covering about 5 million image-text pairs, and completes in around 10 hours on 4 A100 (80GB) GPUs.

**Issues of the first pretraining stage** After its initial pretraining, MiniGPT-4 shows the ability to hold a wealth of knowledge and respond reasonably to human queries. Yet, it sometimes generates incoherent outputs like repetitive words or sentences, fragmented phrases, or irrelevant content, which impairs its capacity for fluent visual conversation with humans.

GPT-3, despite its extensive language dataset pretraining, faced challenges in aligning outputs with user intentions. Instruction finetuning and reinforcement learning from human feedback transformed it into GPT-3.5 (Ouyang et al., 2022; OpenAI, 2022), enhancing its ability to produce human-friendly outputs. This mirrors MiniGPT-4's state after pretraining, explaining its current difficulties in generating fluent, natural human language outputs.

## 3.2 CURATING A HIGH-QUALITY ALIGNMENT DATASET FOR VISION-LANGUAGE DOMAIN.

To achieve greater naturalness in the generated language and enhance the model's usability, a second-stage alignment process is essential. While in the realm of NLP, instruction fine-tuning datasets (Taori et al., 2023) and conversations (sha, 2023) are easily accessible, no equivalent datasets exist for the vision-language domain at the time of this project. To address this deficiency, we curated a detailed image description dataset, specifically tailored for vision-language alignment purposes. This dataset is subsequently utilized to fine-tune our MiniGPT-4 during the second-stage alignment process.

**Initial aligned image-text generation** In the initial phase, we employ the model derived from the first pretraining stage to generate comprehensive descriptions of input images. To enable our model to produce more detailed image descriptions, we designed a prompt that adheres to the conversational format of the Vicuna (Chiang et al., 2023) language model, as shown below. In this prompt, *<ImageFeature>* represents the visual features produced by the linear projection layer.

*###Human: <ImageFeature></Img>Describe this image in detail. Give as many details as possible. Say everything you see. ###Assistant:*

To identify incomplete sentences, we examine whether the generated sentence exceeds 80 tokens. If it does not, we incorporate an additional prompt, *###Human: Continue ###Assistant:* , prompting our MiniGPT-4 to extend the generation process. By concatenating the outputs from both steps, we can create a more comprehensive image description. This approach enables us to generate image-text pairs with detailed and informative image descriptions. We randomly select 5,000 images from the Conceptual Caption dataset (Changpinyo et al., 2021; Sharma et al., 2018) and use the pretrained model to generate corresponding language descriptions for each image.

**Data post-processing** The generated image descriptions are marred by issues like repetitive words or sentences, fragmented sentences, and irrelevant content. To rectify these, we use ChatGPT with a specific prompt to improve the descriptions.

*Fix the error in the given paragraph. Remove any repeating sentences, meaningless characters, not English sentences, and so on. Remove unnecessary repetition. Rewrite any incomplete sentences. Return directly the results without explanation. Return directly the input paragraph if it is already correct without explanation.*

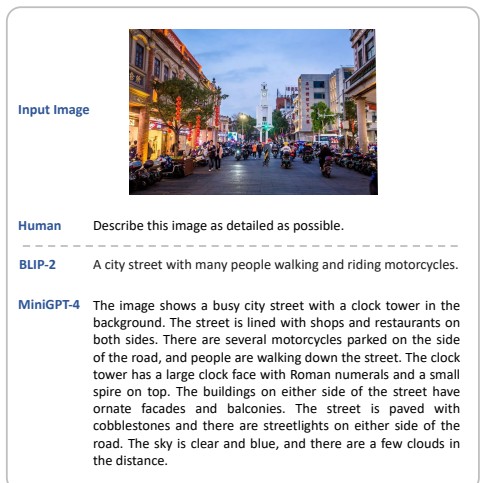

Figure 2: Detailed description

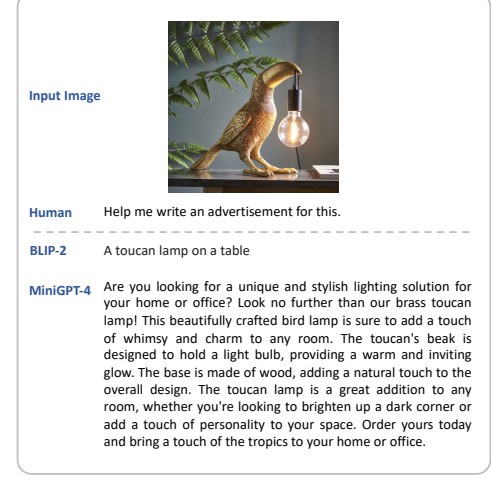

Figure 3: Advertisement promotion

Upon completing the post-processing stage, we manually verify the correctness of each image description to guarantee its high quality. Specifically, we first identified several frequently shown errors (*"I'm sorry I made a mistake...", or "I apologize for that ..."*) and then hard-coded rules to automatically filter them out. We also manually refine the generated captions by eliminating redundant words or sentences that ChatGPT fails to detect. Finally, only approximately 3,500 out of 5,000 image-text pairs satisfy our requirement, and these pairs are subsequently utilized for the second-stage alignment process.

### 3.3 Second-stage finetuning

During the second stage, we finetune our pretrained model with the curated high-quality image-text pairs. During the finetuning, we use the predefined prompts in the following template:

*###Human: <ImageFeature></Img><Instruction>###Assistant:*

In this prompt, *<Instruction>* represents a randomly sampled instruction from our predefined instruction set containing variant forms of instructions such as *"Describe this image in detail"* or *"Could you describe the contents of this image for me"*. It is important to note that we do not calculate the regression loss for this specific text-image prompt.

As a result, MiniGPT-4 is now capable of producing more natural and reliable language outputs. Furthermore, we observed that this fine-tuning process is remarkably efficient, only requiring a mere 400 training steps with a batch size of 12, which takes around 7 minutes with a single A100 GPU.

## 4 Experiments

In the experiment, we aim to showcase the diverse and emergent capabilities of our MiniGPT-4 model through various qualitative examples. These abilities include generating detailed image descriptions, identifying amusing aspects within memes, providing food recipes from photos, writing poems for images, etc. Additionally, we present quantitative results on the task of image captioning.

### 4.1 Uncovering emergent abilities with MiniGPT-4 through qualitative examples

MiniGPT-4 demonstrates many advanced abilities compared to traditional vision-language models. For example, it can describe images in detail and interpret the humorous aspects of a given meme. Here, we qualitatively compared our model to one of the leading vision-language models, BLIP-2 (Li et al., 2023c), with eight distinct examples, each highlighting a different ability.

Fig.2 shows MiniGPT-4's ability to identify multiple elements in an image, like busy streets, clock towers, shops, streetlights, and restaurants, whereas BLIP-2 only notes streets, people, and motorcycles. In another instance, Fig.4a, MiniGPT-4 aptly explains the humor in a meme by relating the dog's expression to common Monday blues, a concept BLIP-2 misses, merely describing the image without grasping its humorous aspect.

MiniGPT-4 has many other capabilities, including creating ads from images (Fig.3), extracting facts from movie photos (Fig.8), generating recipes from food images (Fig.11), diagnosing and suggesting treatments for plant diseases (Fig.12), designing websites from hand-written drafts (Fig.4b), and writing poems inspired by images (Fig.10). These abilities surpass those of traditional models like BLIP-2, which uses Flan-T5 XXL (Chung et al., 2022) as a language model. This difference highlights the importance of aligning visual features with an advanced LLM like Vicuna (Chiang et al., 2023) to unlock advanced vision-language capabilities.

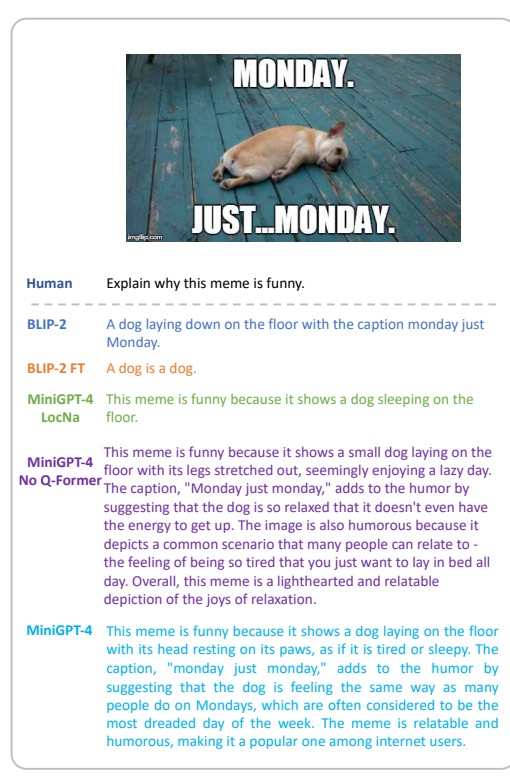

(a) Meme explaining        (b) Website Creating

Figure 4: Model generations from BLIP-2, BLIP-2 finetuned our second stage data (BLIP-2 FT), MiniGPT-4 finetuned with Local Narrative data in the second stage (MiniGPT-4 LocNa), MiniGPT-4 model without Q-Former (MiniGPT-4 No Q-Former), and MiniGPT-4.

Table 1: Quantitative results on advanced vision-language tasks. MiniGPT-4 shows strong performance and successfully responses to 65% of the requests.

|          | Meme  | Recipes | Ads   | Poem  | Avg.   |
|----------|-------|---------|-------|-------|--------|
| BLIP-2   | 0/25  | 4/25    | 1/25  | 0/25  | 5/100  |
| MiniGPT-4 | 8/25  | 18/25   | 19/25 | 20/25 | 65/100 |

## 4.2 QUANTITATIVE ANALYSIS

**Advanced Abilities**   Our evaluation dataset for vision-language tasks included 100 images divided across four tasks: meme interpretation, recipe generation, advertisement creation, and poem composition, each with 25 images. Human evaluators assessed the model's responses. We compared MiniGPT-4 with BLIP-2, as detailed in Tab.1. MiniGPT-4 outperformed BLIP-2 (Li et al., 2023c), especially in recipe, advertisement, and poem tasks, successfully handling 80% of these. It also interpreted humor in memes correctly in 8 out of 25 cases, a challenging aspect for BLIP-2.

**Image Captioning**   We evaluate the performance of MiniGPT-4 on the COCO caption benchmark and compare it with BLIP-2 (Li et al., 2023c). Our model's generated captions typically contain rich visual details. As such, conventional similarity-based image-caption evaluation metrics struggle to provide an accurate evaluation. To evaluate, we check how many of COCO's 5 ground truth captions per image are covered by MiniGPT-4's captions, using GPT-4 turbo. Evaluation details can be found in Appx.A.3. Results in Tab.2 show MiniGPT-4 averaged 2.22 ground truth captions, better than BLIP-2's 1.96, proving its captions to be more informative. Additional evaluations on traditional VQA tasks are detailed in Appx.A.2.

**Video Understanding** Here, we evaluate MiniGPT-4 for video understanding. We finetuned MiniGPT-4 on 1.2k videos from the VideoInstruct100K (Maaz et al., 2023), using 50 frames and subtitles per video. Experimental results on the video-based generative performance benchmark  (Maaz et al., 2023) in Tab. 4 show that MiniGPT-4 outperformed the strongest baseline Video-ChatGPT  (Maaz et al., 2023) in correctness, detail, context, and time comprehension, while also showing strong consistency, demonstrating MiniGPT-4's potential in processing videos.

**Other Benchmarks** MinGPT-4 has been densely evaluated and compared with contemporary baselines like LLaVa (Liu et al., 2023a) and mPlug-Owl (Ye et al., 2023) by many popular benchmarks like MMBench (Liu et al., 2023b) quantitatively. A detailed discussion of MiniGPT-4's performance on these benchmarks can be found in Appx.A.5.

## 4.3 ANALYSIS ON THE SECOND-STAGE FINETUNING

**Effectiveness of the second-stage finetuning**   Utilizing MiniGPT-4 solely after the first pretraining stage leads to issues like repetitive or fragmented sentences. These are largely resolved after the second-stage finetuning, as shown in Fig.5, where MiniGPT-4 evolves from generating incomplete to fluent captions. This section assesses the second-stage finetuning's importance and effectiveness.

To measure its impact, we sampled 100 images from the COCO test set for the detailed description and poem writing tasks, using the prompts "*Describe the image in detail.*" and "*Can you write a beautiful poem about this image?*". Both pre- and post-second-stage finetuned models attempted these tasks. Results in Tab.3 show a significant drop in failures post-finetuning, with less than two failures

Table 2: COCO caption evaluation. We use GPT-4 turbo to count the number of ground truth captions the model output can cover. MiniGPT-4(GPT-4v) denotes a variant trained using GPT-4V generated data in the second stage.

|  | BLIP-2 | MiniGPT-4 | MiniGPT-4 (GPT-4v) |
|---|---|---|---|
| #GT Cover | 1.96 | 2.22 | 2.26 |

Table 3: Failure rates of detailed caption and poem generation tasks before and after second-stage finetuning. The finetuning stage significantly reduces generation failures.

| Failure rate | Detailed caption | Poem |
|---|---|---|
| Before stage-2 | 35% | 32% |
| After stage-2 | 2% | 1% |

|  | Correctness | Detail | Contextual | Temporal | Consistency |
|---|---|---|---|---|---|
| Video Chat (Li et al., 2023d) | 2.23 | 2.50 | 2.53 | 1.94 | 2.24 |
| Llama Adapter (Zhang et al., 2023b) | 2.03 | 2.32 | 2.30 | 1.98 | 2.15 |
| Video LLama (Zhang et al., 2023a) | 1.96 | 2.18 | 2.16 | 1.82 | 1.79 |
| Video-ChatGPT (Maaz et al., 2023) | 2.40 | 2.52 | 2.62 | 1.98 | **2.37** |
| MiniGPT-4 | **2.68** | **2.76** | **3.20** | **2.26** | 2.18 |

Table 4: Video understanding on the video-based generative performance benchmark.

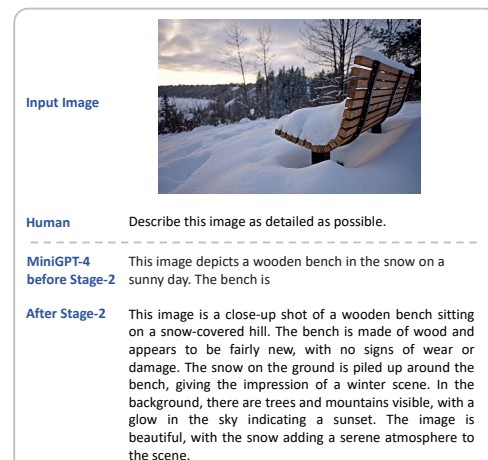
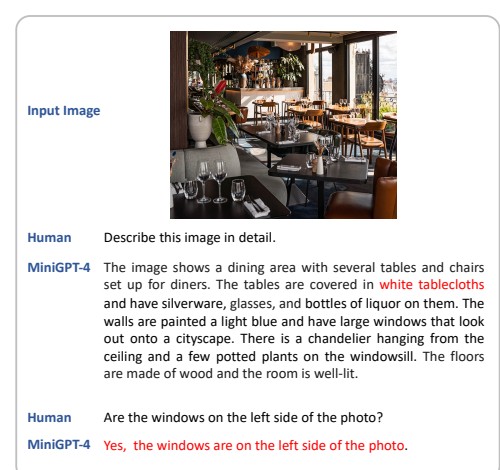

Figure 5: MiniGPT-4 before second-stage fine-tuning fails to output completed texts. The generation is improved after the finetuning.

Figure 6: An example of MiniGPT-4's limitations. MiniGPT-4 hallucinates unexisting tablecloths and can't locate the windows correctly.

in 100 images for each task, indicating a notable improvement in output quality. Fig.5 provides qualitative examples of this enhancement.

**Can the original BLIP-2 benefit from the second-stage data?** In this study, we finetune BLIP-2 (Li et al., 2023c) with our second-stage data in the same way as MiniGPT-4, and check if it can obtain similar advanced abilities as MiniGPT-4. The finetuned BLIP-2 is denoted as BLIP-2 FT. Note that MiniGPT-4 uses the same visual module as BLIP-2; while BLIP-2 uses FlanT5 XXL (Chung et al., 2022) as the language model, which is not as strong as the Vicuna (Chiang et al., 2023) model used in our MiniGPT-4 model. We rely on the same prompts to assess the advanced capabilities of our model. Qualitative results are shown in Fig.4, 13, and 14. We discover that BLIP-2 FT still generates short responses and fails to generalize to advanced tasks like meme explaining and website coding (Fig.4). Our finding suggests that BLIP-2's relatively weaker language model FlanT5 XXL benefits less from such a small dataset, and highlights the effectiveness of a more advanced LLM in a VLM system.

**Second stage with Localized Narratives** We tested MiniGPT-4's performance by substituting our self-collected dataset with the Localized Narratives dataset (Pont-Tuset et al., 2020) in the second training stage. We name this variant MiniGPT-4 LocNa. The Localized Narratives dataset features detailed image descriptions with corresponding regional localizations. Qualitative results shown in Fig.4, 13, and 14 reveal that MiniGPT-4 LocNa can produce lengthy image descriptions (as seen in Fig.14). However, these outputs are of lower quality, often with monotonous expressions. MiniGPT-4 LocNa also shows weaker generalization in complex tasks, like explaining meme humor (Fig.4a), compared to the original MiniGPT-4. This performance difference may stem from the repetitive and monotonous nature of the Localized Narratives dataset.

**Second stage with GPT-4V generated data.** We conduct further ablation experiments using 2,000 GPT-4V generated image-text pairs collected by LAION (LAION, 2023) in the second stage. Results in Tab.2 shows performance improvements from this fine-tuning.

Table 5: Ablation on architecture designs

| Model | AOK-VQA | GQA |
|---|---|---|
| MiniGPT-4 | 58.2 | 32.2 |
| (a) MiniGPT-4 w/o Q-Former | 56.9 | 33.4 |
| (b) MiniGPT-4 + 3 Layers | 49.7 | 31.0 |
| (c) MiniGPT-4 + Finetune Q-Former | 52.1 | 28.0 |

Table 6: Hallucination Evaluation

| | CHAIR$_i$ | Avg. Length |
|---|---|---|
| Blip-2 | 1.3 | 6.5 |
| mPLUG-Owl | 30.2 | 98.5 |
| LLaVa | 18.8 | 90.7 |
| MiniGPT-4 (short) | 7.2 | 28.8 |
| MiniGPT-4 (long) | 9.6 | 175 |

**Amount of traing data in the first stage**    This ablation study can be found in Appx.A.4.

## 4.4    ABLATION ON THE ARCHITECTURE DESIGNS

To further demonstrate the effectiveness of using one single linear layer to align visual features with LLM, we conduct experiments with different architecture designs, including (a) removing the Q-Former and directly mapping the VIT's output to Vicuna's embedding space (i.e., without Q-former), (b) using three linear layers instead of one layer, and (c) additionally finetuning the Q-Former in the vision module. All the variants are trained in the same way as the original design. Results on AOK-VQA (Schwenk et al., 2022) and GQA (Hudson & Manning, 2019) datasets in Tab.5 show that the variant (a) **MiniGPT-4 w/o Q-Former** has a similar performance to the original design. Qualitative results of this variant in Fig.4, 13, and 14 also show similar advanced skills. This reveals that the Q-Former from BLIP-2 doesn't plays a critical roles for advanced skills. Besides, both variants (b) **MiniGPT-4+ 3 Layers** and (c) **MiniGPT-4 + finetuning Q-Former**, perform slightly worse than the original MiniGPT-4. This indicates a single projection layer is sufficient to align the vision encoder and the large language model in our limited training data setting.

## 4.5    LIMITATION ANALYSIS

**Hallucination**    As MiniGPT-4 is built upon LLMs, it inherits LLM's limitations like hallucinating nonexistent knowledge. An example in Fig. 6 shows that MiniGPT-4 incorrectly identifies the presence of white tablecloths in the image, despite their absence. Here, we use the metric $CHAIR_i$ (Rohrbach et al., 2018) to gauge the hallucination rate of the generation, with the two distinct prompts to control the model generation length: *MiniGPT-4 (long)*: Please describe this image as detailed as possible. *MiniGPT-4 (short)*: Please describe the image shortly and precisely, in less than 20 words.

Results in Tab.6 show that longer captions tend to have higher hallucination rates. For example, MiniGPT-4 (long) generates captions averaging 175 words with a higher hallucination rate, while MiniGPT-4 (short) averages 28.8 words with a lower rate. BLIP-2, averaging 6.5 words, hallucinates less but covers fewer objects as seen in Tab.2. Compared to contemporary methods like LLaVa or mPlug-Owl, MiniGPT-4 generates longer descriptions with fewer hallucination. Hallucination in detailed image descriptions is still an unresolved issue. Using Reinforcement Learning with AI feedback with hallucination detection modules may be a potential solution.

**Spatial Information Understanding**    MiniGPT-4's visual perception remains limited. It may struggle to differentiate spatial localization. For example, MiniGPT-4 in Fig. 6 fails to identify the location of the windows. This limitation may stem from a lack of aligned image-text data designed for spatial information understanding. Training on such datasets like RefCOCO (Kazemzadeh et al., 2014) or Visual Genome (Krishna et al., 2017) could potentially alleviate this issue.

## 5    DISCUSSION

How does MiniGPT-4 obtain these advanced abilities? Many of the advanced vision-language capabilities demonstrated by GPT-4 can be understood as compositional skills rooted in two foundational skills: image understanding and language generation. Take the task of image-based poem writing as an example. Advanced LLMs like ChatGPT and Vicuna can already craft poems based on users' instructions. If they acquire the ability to understand images, compositionally generalizing to the task of image-based poem writing even without having image-poem pairs in their training data is possible.

In its first pretraining stage, MiniGPT-4 learns image understanding by correlating images with short descriptions from caption datasets. However, the language style in these datasets differs from that of modern LLMs, leading to distorted language generation and impeding compositional generalization. To address this, a second-stage finetuning is introduced to improve language generation. Post two-stage training, MiniGPT-4 successfully demonstrates advanced compositional vision-language abilities, such as draft-to-website or interpreting memes, confirming our approach. Future research could explore the mechanisms of compositional generalization further. Our work, as a preliminary exploration of vision-based LLM capabilities, aims to encourage more studies in this area.

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

# A   APPENDIX

## A.1   MORE QUALITATIVE RESULTS

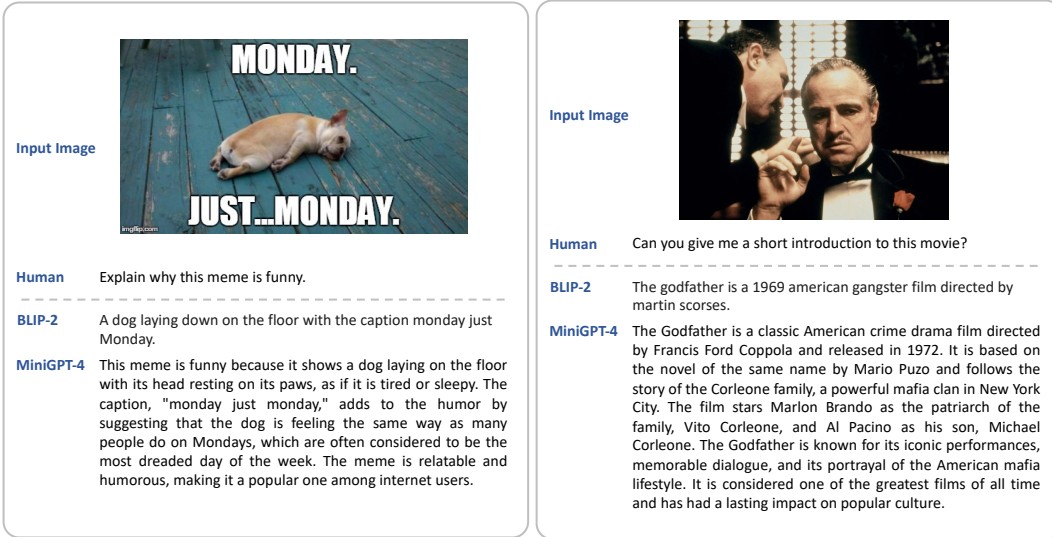

Figure 7: Advertisement promotion

Figure 8: Factual retrieval

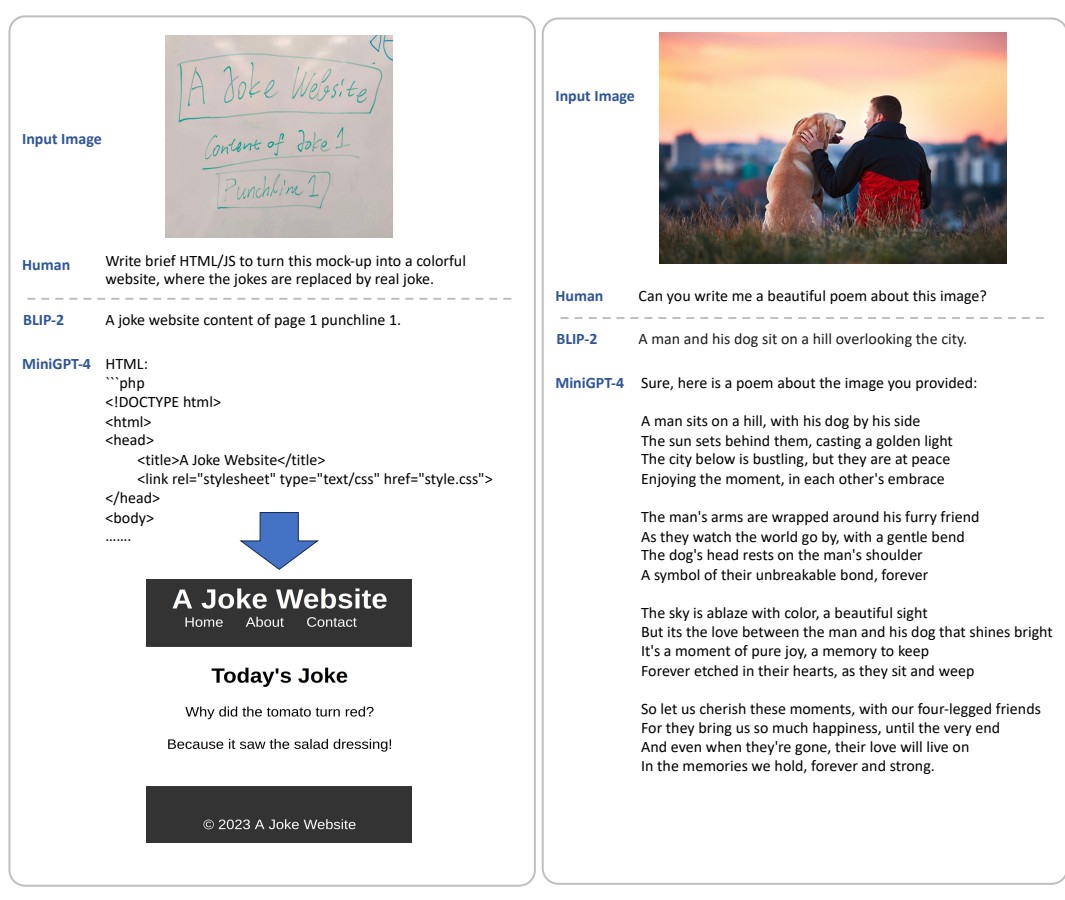

Figure 9: Website creating

Figure 10: Poem writing

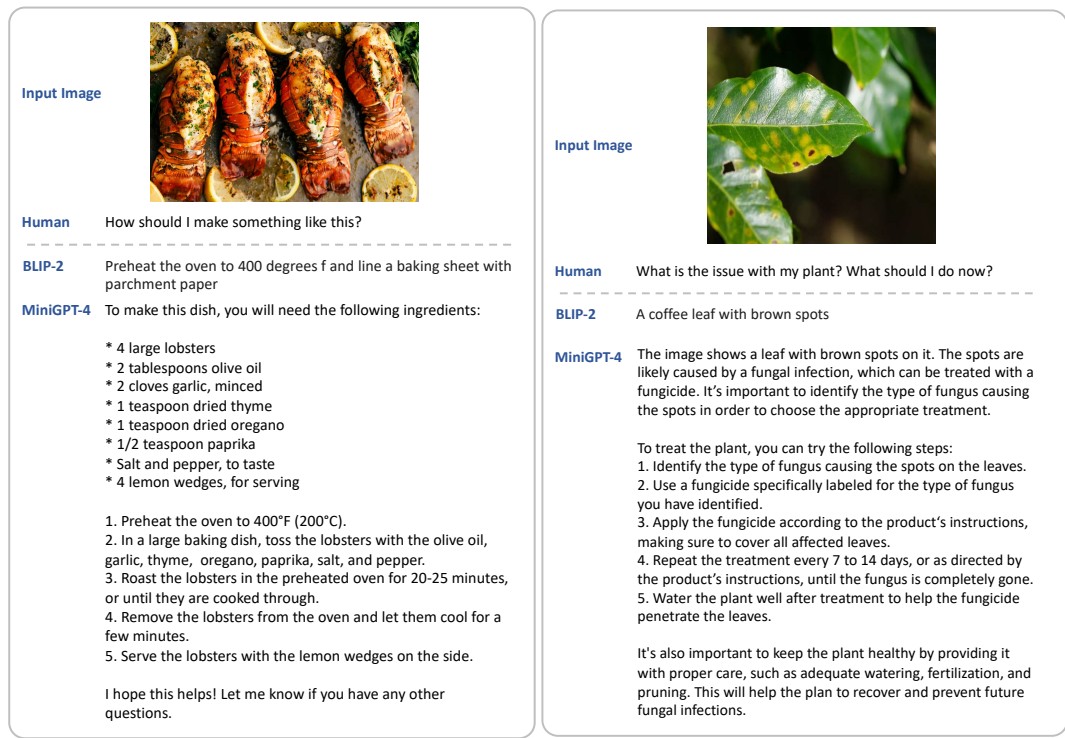

Figure 11: Food recipe generation

Figure 12: Plant cultivating

## A.2 EVALUATION IN TRADITIONAL VQA BENCHMARKS

The aim of this study is to replicate the remarkable multi-modal capabilities demonstrated in GPT-4, such as generating detailed image descriptions and creating websites from hand-drawn drafts. To emphasize the most crucial component of advanced vision-language skills, the methodology of MiniGPT-4 is intentionally kept minimal. For instance, the learnable model capacity is limited (only one linear layer), and MiniGPT-4 is trained with just 5 million pairs, in contrast to BLIP-2 with 129 million image-text pairs. Such a pared-down approach is anticipated to yield suboptimal results on traditional benchmarks. While this isn't our primary goal, we offer a quantitative analysis of the VQA datasets A-OKVQA (multi-choice) (Schwenk et al., 2022) and GQA (Hudson & Manning, 2019). Additionally, to showcase the potential of MiniGPT-4 with traditional benchmarks, we conduct a straightforward ablation study. Here, we simply unfreeze the LLM using LoRA (Hu et al., 2021) and incorporate more training data from the VQAv2, OKVQA, and A-OKVQA datasets during the second finetuning stage. Results in Tab. 7 indicate that the original MiniGPT-4 lags behind BLIP-2 by a reasonable margin, and merely augmenting the learning capacity and the training data results in a substantial performance improvement, which confirms our expectations. We believe our model's performance on conventional vision benchmarks can be enhanced with a carefully designed training strategy (e.g., dataset sample ratios, learning rate schedule, etc.), more training data/datasets, and additional learnable parameters. Since enhancing performance on traditional vision benchmarks isn't this project's objective, we reserve this aspect for future research.

| Model | Training data | AOK-VQA | GQA |
|---|---|---|---|
| Blip-2 | 129M image-text pairs | 80.2 | 42.4 |
| MiniGPT-4 | 5M image-text pairs | 58.2 | 32.2 |
| MiniGPT-4 (Finetune Vicuna) | 5M image-text pairs | 67.2 | 43.5 |

Table 7: Performance Comparison between BLIP-2 and MiniGPT-4

## A.3 DETAILS OF CAPTION EVALUATION

We utilize GPT-4 turbo (gpt-4-1106-preview) to assess whether the generated descriptions capture the content of each ground truth caption individually. In the COCO dataset, each image is accompanied by 5 ground truth captions. For every image, we calculate the number of captions covered by the generated descriptions and then average this count across 5000 random sampled images from the validation set to derive the final score.

Here is the prompt we use in GPT-4 turbo
*Given a test image description and a list of gt image caption,*
*verify whether the information in gt caption is included in the test description.*
*The answer should be yes or no.*
*Input is in this format:*
*Test: (test sentence)*
*1: (gt1)*
*2: (gt2)*
*3: (gt3)*
*you need to answer yes or no for each gt in the following format:*
*1: (yes/no)*
*2: (yes/no)*
*3: (yes/no)*

## A.4 AMOUNT OF TRAINING DATA IN THE FIRST STAGE.

We evaluate the impact of training data volume in the first stage by using checkpoints at 10%, 30%, and 50% of stage 1 duration, subsequently finetuned in stage 2. Results in Tab. 8 show a significant performance drop with only 10% of stage 1 data. However, utilizing 30% of stage 1 data, equivalent to 1.5M image-text pairs can achieve similar performance with the original MiniGPT-4. No gains were seen beyond 50% of stage 1 data, indicating potential saturation of the model's learnable capacity at this juncture.

| Metric | 10% | 30% | 50% | 100% |
|---|---|---|---|---|
| #GT Cover | 1.62 | 2.15 | 2.26 | 2.22 |

Table 8: Captioning performance with different amount of training data in stage-1.

## A.5 MINIGPT-4 ON MMBENCH

MMBench (Liu et al., 2023b) is a new multi-modality benchmark with diverse evaluation questions to evaluate different abilities of vision language model. MMBench evaluated MiniGPT-4 together with other contemporary vision language models like OpenFlamingo (Awadalla et al., 2023), VisualGLM (Du et al., 2022), LLaVa (Liu et al., 2023a), and InstructBlip Dai et al. (2023). Here, we show the performance of MiniGPT-4 and other baseline models in Tab. 9. Results show that MiniGPT-4 demonstrates competitive performance compared to contemporary methods, e.g., InstructBlip. It surpasses InstructBlip in several key areas: logical reasoning (LR), fine-grained perception for single instance (FP-S), and fine-grained perception across instances (FP-C). Additionally, MiniGPT-4 achieves competitive results in relation reasoning (RR), attribute reasoning (AR), and coarse perception (CP).

| Model | Overall | LR | AR | RR | FP-S | FP-C | CP |
|-------|---------|-----|------|------|------|------|------|
| OpenFlamingo | 4.6 | 6.7 | 8.0 | 0.0 | 6.7 | 2.8 | 2.0 |
| VisualGLM | 38.1 | 10.8 | **44.3** | 35.7 | 43.8 | 23.4 | 47.3 |
| LLaVa | 38.7 | 16.7 | 48.3 | 30.4 | 45.5 | 32.4 | 40.6 |
| InstructBlip | **44.0** | 19.1 | **54.2** | 34.8 | 47.8 | 24.8 | **56.4** |
| MiniGPT-4 | 42.3 | **20.8** | 50.7 | 30.4 | **49.5** | **26.2** | 50.7 |

Table 9: Perforance on MMBench benchmark. Numbers are from Liu et al. (2023b).

## A.6 MORE QUALITATIVE ABLATION RESULTS

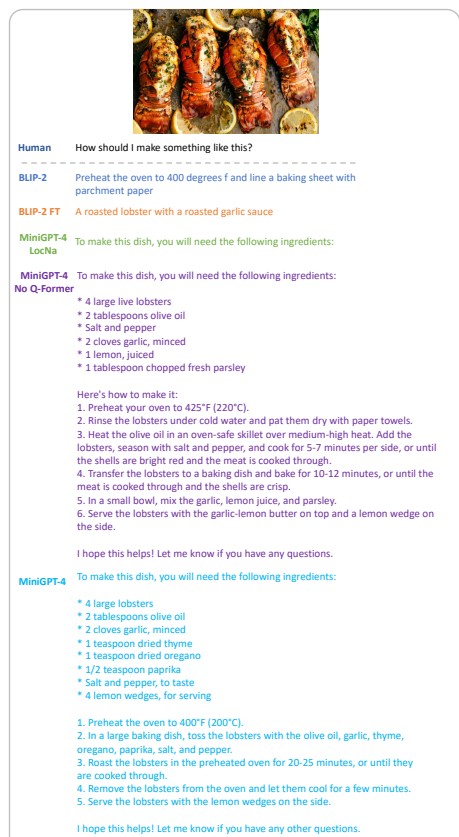

Figure 13: Ablation Study on Recipe Generation

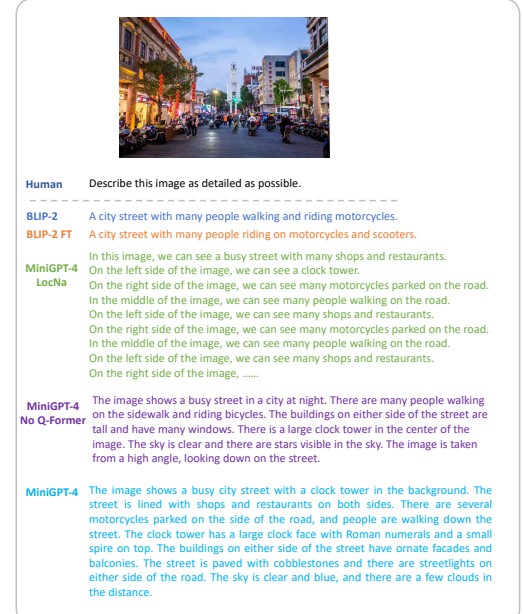

Figure 14: Ablation Study on Detailed Description

