# OpenReview forum: "MiniGPT-4: Enhancing Vision-Language Understanding with Advanced Large Language Models"
_ICLR.cc/2024/Conference — ICLR 2024 poster_

### Official Review · Reviewer_auSX · 2023-10-15

**Soundness:** 3 good
**Presentation:** 3 good
**Contribution:** 2 fair
**Rating:** 6
**Confidence:** 4

**Summary:**

This paper presents a recipe for reproducing the GPT-4 vision model. The approach involves aligning the pre-trained vision encoder of BLIP-2 (ViT + Qformer) with the LLM Vicuna using a noisy image-caption dataset. The model is then fine-tuned with a curated detailed image description dataset, which is generated by refining the initial model predictions using ChatGPT and manual refinement. Experimental results demonstrate the capabilities of the proposed MiniGPT-4 model in challenging vision-language tasks like Poem Generation and Meme understanding. While qualitative results are mostly presented, quantitative findings indicate that MiniGPT-4 outperforms BLIP-2 on image captioning when evaluated with ChatGPT, although it tends to generate more hallucinations than BLIP-2.

**Strengths:**

- The exploration to achieve a GPT-4 level VLMs by combining LLMs & vision encoders is greatly appreciated. Additionally, the provision of open-sourced model checkpoints and dataset enhances the reproducibility and transparency of the research.
- The qualitative results presented in the study demonstrate the potential effectiveness of the proposed recipe. The process of aligning the model through training on a noisy dataset and subsequently fine-tuning it with a high-quality dataset shows promise in improving the model's performance and response quality.

**Weaknesses:**

- The quantitative evaluation setup lacks convincing elements:
  - The paper only compares the proposed model with BLIP-2 as the baseline, which utilizes a FLAN-T5 model as the base LLM. To enhance the credibility of the results, it would be beneficial to compare the proposed model with more recent open-sourced components that utilize similar base LLMs, such as LLaVA[1], InstructBLIP[2], and mPlug-Owl[3].
  - The selection of the four Advanced Abilities tasks appears arbitrary, and the evaluation sets for these tasks are relatively small (25 samples for each task). To provide a more comprehensive evaluation, it is recommended to include evaluation on recent multimedia benchmarks such as TouchStone[4] and MMBench[5].
  - In Table 2, the coco caption evaluation with ChatGPT seems questionable. The approach of randomly selecting a ground-truth caption and asking ChatGPT if the generated text covers the main visual concepts in the label text may lead to instances where the selected "ground-truth" caption and the generated text depict the image in different aspects, yet both are considered correct. Reporting standard metrics such as BLEU and Rouge scores, which consider multiple candidate ground-truth captions, could mitigate the issues.
  - Table 5 lacks the CHAIR_S score for different methods. Additionally, it indicates that MiniGPT-4 fails to follow the instruction's requirement of 20 words, as evidenced by the average length of 27 words for MiniGPT-4 (short). To me, instead of showcasing the promising results on so-called adavanced tasks, satisfying the user requirement of a basical instruction should be of higher priority.
- Some claims made in the paper are misleading:
  - In Section 3.2, the paper claims that no equivalent datasets exist (for multi-modal instruction tuning) in the vision-language domain. However, there have been recent studies addressing this problem, such as LLaVA[1], InstructBLIP[2], M3IT[6], and MultiInstruct[7]. To accurately reflect the current state of the field, the author should properly cite these studies and consider exploring further fine-tuning with these datasets.
  - In Section 4.4, the paper claims that MiniGPT-4 achieves similar results with Qformer, implying that Qformer does not play a critical role in advanced skills. However, this claim may not be fully supported by the AOK-VQA and GQA results, along with additional case studies. Additionally, if the advanced skills require inference between multiple images, the downsampling effect of Qformer (i.e., downsampling visual patches from 256 to 32) could be crucial and should be considered.

## References

It is not a curated list, and most of them appeared  more than 3 months before the submission deadline.

[1] Visual Instruction Tuning, https://arxiv.org/abs/2304.08485

[2] InstructBLIP: Towards general-purpose vision-language models with instruction tuning, https://arxiv.org/abs/2305.06500

[3] mPLUG-Owl: Modularization Empowers Large Language Models with Multimodality, https://arxiv.org/abs/2304.14178

[4] TorchStone: https://github.com/QwenLM/Qwen-VL/blob/master/touchstone/README.md

[5] MMBench: https://opencompass.org.cn/leaderboard-multimodal

[6] M3IT: A Large-Scale Dataset towards Multi-Modal Multilingual Instruction Tuning, https://arxiv.org/abs/2306.04387

[7] MultiInstruct: Improving Multi-Modal Zero-Shot Learning via Instruction Tuning, https://arxiv.org/abs/2212.10773

**Questions:**

Q1: What criteria did you use to select the Advanced Abilities tasks? Additionally, can you clarify if the images used for evaluation in these tasks overlap with the dataset used in stage 1 pre-training and stage 2 fine-tuning?

Q2: Could you specify the version of ChatGPT that you adopted for answer revision in your experiments?

---

> ### Author Response · Authors · 2023-11-21
> **Response**
>
> Thank you for your valuable feedback! We response to the questions as follows:
>
> **Q1 It would be beneficial to compare the proposed model with more recent open-sourced components that utilize similar base LLMs, such as LLaVA, InstructBLIP, and mPlug-Owl. It is recommended to include evaluation on recent multimedia benchmarks such as MMBench.**
>
> As mentioned by Reviewer 7oDs, MiniGPT4 has been evaluated in the MMBench benchmark paper, and also in other benchmarks like MME and VisIT-Bench, together with similar baseline methods like LLaVa and InstructBLIP.  Please refer to Q0 in the general response for a detailed discussion.
>
> **Q2 What criteria did you use to select the Advanced Abilities tasks? Additionally, can you clarify if the images used for evaluation in these tasks overlap with the dataset used in stage 1 pre-training and stage 2 fine-tuning?**
>
> For our study, we chose advanced abilities that necessitate both an understanding of visual content and complex language generation. We gathered the images for our dataset using Google Image Search, employing related keywords such as "funny memes" and "photos of delicious food."
>
> Furthermore, it's important to note that the training set consists solely of image description texts and does not include any text data pertaining to advanced abilities. MiniGPT4 was not trained with image-text paired data to learn these skills. However, it gains this skill compositionally by efficiently mapping images to the embedding space that the LLM (frozen by default) can operate on.
>
> **Q3 In Table 2, the approach of randomly selecting a ground-truth caption and asking ChatGPT if the generated text covers the main visual concepts in the label text may lead to instances where the selected "ground-truth" caption and the generated text depict the image in different aspects, yet both are considered correct. Reporting standard metrics such as BLEU and Rouge scores, which consider multiple candidate ground-truth captions, could mitigate the issues. Could you specify the version of ChatGPT that you adopted for answer revision in your experiments?**
>
> In Table 2, we focus not on the correctness of the generated captions but on their information coverage. A high score indicates that the generated descriptions encompass more information from the image, rather than “more correct”. For this evaluation, we employed the API gpt-3.5-turbo.
>
> Traditional metrics such as BLEU and Rouge, which rely on sentence similarity, fall short when the format of evaluated image descriptions diverges from the ground truth. This discrepancy is particularly noticeable when the generated descriptions are significantly longer than the original ground truth captions. We designed a new metric based on GPT-4 that takes all the ground truth captions into account. Please refer to Q2 in the general response section for details.
>
> **Q4 Table 5 lacks the CHAIR score for different methods.**
>
> Kindly refer to Q5 in the general response where we include results from more methods.
>
> **Q5 MiniGPT-4 fails to follow the instruction's requirement of 20 words, as evidenced by the average length of 27 words for MiniGPT-4 (short). To me, satisfying the user requirement of a basic instruction should be of higher priority.**
>
> The ability to generate a sentence with exactly 20 words is constrained by the LLM's capabilities. While SOTA closed-sourced models like GPT-4 perform well in this task, open-source models like Llama2 13B struggle. We conducted a small experiment with Llama2 13B, instructing it to "generate a sentence with 20 words". The average number of words over the 50 generations is only 14.8 words. This seemingly simple instruction proves challenging for a 13B open-source LLM. Since we freeze the open-source LLM on MiniGPT-4, it's unsurprising that MiniGPT-4 can't precisely generate a 20-word caption. Nonetheless, this prompt still encourages MiniGPT-4 to significantly reduce the image description length from 175 to 28.8 words
>
> **Q6 In Section 3.2, the paper claims that no equivalent datasets exist (for multi-modal instruction tuning) in the vision-language domain. However, there have been recent studies addressing this problem, such as LLaVA, InstructBLIP, M3IT, and MultiInstruct. To accurately reflect the current state of the field, the author should properly cite these studies and consider exploring further fine-tuning with these datasets.**
>
> At the time when this project was developed, there was no other multi-modal instruction tuning dataset that existed. We updated section 3.2 (highlighted in blue) in the updated paper to better reflect this background. In addition, we updated the related work section to better represent the current landscape.
> Furthermore, we carried out a new experiment, substituting our second-stage dataset with a newly-released image description dataset from LAION. This dataset features descriptions generated by GPT-4v. For detailed information, please refer to Q3 in the general response section.

---

> > ### Author Response · Authors · 2023-11-21
> > **Response Part II**
> >
> > **Q7 If the advanced skills require inference between multiple images, the downsampling effect of Qformer could be crucial and should be considered.**
> >
> > Exploring the downsampling effect indeed presents an intriguing avenue for future research. For the possible multi-image applications, we have fine-tuned a video version of MiniGPT-4. For more comprehensive details, please refer to Q1 in the general response section.
> >
> >
> > **Q8 In Section 4.4, the paper claims that MiniGPT-4 achieves similar results with Qformer, implying that Qformer does not play a critical role in advanced skills. However, this claim may not be fully supported by the AOK-VQA and GQA results, along with additional case studies.**
> >
> > The variant of our model that excludes the Q-Former demonstrates similar performance to the original version in the AOK-VQA and GQA results. Specifically, it achieved a score 1.2 points higher than the original MiniGPT-4 in GQA and a score 1.3 points lower in AOK-VQA, as detailed in Table 4 of Section 4.4. Additional qualitative analysis presented in Figures 4, 13, and 14 also reveals comparable results between the no-Q-former variant and the original model. Based on these findings, we believe our results substantiate our claim.

---

> > > ### Comment · Reviewer_auSX · 2023-11-22
> > >
> > > Thanks for the detailed response, and most of my concerns are addressed (e.g., more baseline comparison on more benchmarks and the state of current progress). Accordingly, I will raise my score to 6.
> > >
> > > Furthermore, I would like to discuss more about the instruction following ability. I know the difficulty of asking the model to generate a response with the **exact word number**, such as 20 words in the short instruction. Nevertheless, LLaMa-13B could understand the key idea is a short response within 20 words (e.g., 14.8 words), why does the frozen LLM in MiniGPT-4 still generate longer responses (e.g., 28.8 words on average)? Is there a tendency to produce a longer response after fine-tuning and what are the potential side effects?

---

> ### Author Response · Authors · 2023-11-22
>
> Thank you for your comment! To verify more directly, we conducted a simple experiment with Vicuna-13B, the LLM used by MiniGPT-4. We employed the prompt "Introduce me an animal in 20 words." Across 20 samples, the average length of the responses generated was 29.6 words, similar to the average length (28.8 words) in MiniGPT-4's generation from our ablation study. These results suggest that the over generating issue is due to the limitations of MiniGPT-4's LLM.

---

### Official Review · Reviewer_A8a5 · 2023-10-24

**Soundness:** 2 fair
**Presentation:** 3 good
**Contribution:** 3 good
**Rating:** 6
**Confidence:** 4

**Summary:**

The paper introduces MiniGPT-4, which combines a frozen visual encoder with a frozen large language model, Vicuna, using a single projection layer. The research revealed that aligning visual features with a sophisticated large language model can replicate many of GPT-4's advanced multi-modal functions. A two-step training process is proposed, and the second step plays a vital role in generating long and meaningful responses. Additionally, MiniGPT-4 showcased unique capabilities like writing stories or poems inspired by images and teaching users cooking techniques based on food photos.

**Strengths:**

Easy Approach: The paper presents an easy two-step approach to combining a visual encoder with a large language model, providing insights into GPT-4's enhanced multi-modal capabilities.

Diverse Capabilities: MiniGPT-4 replicates many of GPT-4's features, such as writing stories based on images and culinary guidance from food photos.

Interesting Experiments: The authors provide qualitative and quantitative experiments to show the various abilities of the proposed method, including the human study of advanced abilities.

**Weaknesses:**

* The necessity of using a large number of image and text pairs to train a projection layer is not addressed sufficiently.

* The comparisons between BLIP-2 and MiniGPT are unfair because BLIP-2 uses a weaker LLM and is not designed for the advanced abilities mentioned in the paper. Furthermore, the designed new evaluation method using ChatGPT to check whether the generation covers all the objects and visual relations is not compelling. It is possible because many hallucinations are generated, as Table 5 illustrates.

**Questions:**

* Is it possible to use less data from the combined dataset to train the projection layer in the first pertaining stage? It is better to provide insights on the size data necessary to align the vision and text space.

* Surprisingly, the performance drops a lot when three projection layers are used instead of one. Could you try different designs of the projection module, e.g., two or more than three projection layers, or adding activation layers between projection layers to provide insights on why one projection layer is better and why the drop is so huge?

---

> ### Author Response · Authors · 2023-11-21
> **Response**
>
> Thank you for your valuable feedback! We response to the questions as follows:
>
>
> **Q1 The comparisons between BLIP-2 and MiniGPT are unfair because BLIP-2 uses a weaker LLM and is not designed for the advanced abilities mentioned in the paper.**
>
> The essential message our paper conveys to the community is as follows: advanced multimodal capabilities can be attained by utilizing a stronger LLM with advanced pure language skills, even if the LLM is not specifically designed for advanced **multimodal** capabilities. The particularity of our design choices and the details of making it work in a two-stage manner are at the heart of our work/contributions.  In addition, we provide a discussion of MiniGPT-4 in benchmarks like MMBench and MME in our general response. Kindly refer to Q0 of the general response for more details.
>
> **Q2 The designed new evaluation method using ChatGPT to check whether the generation covers all the objects and visual relations is not compelling. It is possible because many hallucinations are generated, as Table 5 illustrates.**
>
>
> From Q5 in the general response, we can see that the hallucination issue of MiniGPT-4 is less obvious compared to other baseline models like LLava or mPLUG-Owl.
> In addition, we introduce a more robust metric that better considers all the ground truth caption into account with a stronger judger GPT-4-turbo. Please refer to Q2 in the general response.
>
>
> **Q3 The necessity of using a large number of image and text pairs to train a projection layer is not addressed sufficiently.**
>
> Kindly refer to Q4 in the general response section, where we respond experimentally to this question.
>
>
> **Q4. The performance drops a lot when three projection layers are used instead of one. Why the drop is so huge?**
>
> The three-layer variant almost triples the learning capacity (the number of learnable parameters). This increase may necessitate more training data compared to the original one-layer version. Consequently, training this variant on the same amount of data as the original version might not suffice to achieve comparable performance.

---

> > ### Comment · Reviewer_A8a5 · 2023-11-22
> >
> > Thank you for your response, which has partially addressed my concerns. Hence, I have decided to maintain my original scores.

---

> ### Author Response · Authors · 2023-11-22
>
> Thanks for your comment. Regarding Q4, our design with a single linear projection layer aligns with previous studies like [1][2]. These works already demonstrated that training only a single linear projection is sufficient for transferring image representations to a frozen LM space. Similarly, another concurrent work, LLaVa, uses a single projection layer to align visual and language features. The significant performance decline observed in AOK-VQA may be attributed to increased training capacities, suggesting the need for more training data than our original single-layer version required. As the reviewer suggests, we will expand our discussion in the final version to include ablations on alternative projection designs and the use of additional alignment data.
>
> [1 ]Merullo, Jack, et al. Linearly Mapping from Image to Text Space. ICLR 2023.
>
> [2] Scialom, Thomas, et al. What BERT sees: Cross-modal transfer for visual question generation. INLG 2020.

---

### Official Review · Reviewer_TFFX · 2023-10-31

**Soundness:** 3 good
**Presentation:** 2 fair
**Contribution:** 4 excellent
**Rating:** 5
**Confidence:** 4

**Summary:**

This paper attempts to create Multimodal LLMs that can understand the visual inputs. To do this, MiniGPT-4 uses a frozen visual encoder along with a frozen vicuna model and only trains the single projection layer. Various emerging properties of the M-LLMs are discussed and a novel detailed image description-based fine-tuning stage is proposed to further improve the image understanding.

Overall this paper has contributed significantly. However, it’s not yet ready for publication. Other rounds of revisions are needed with extensive benchmarking.

**Strengths:**

* One of the first works attempts to introduce visual modalities in pre-trained LLMs in a parameter-efficient manner.
* MiniGPT-4 requires 40 A100 GPU hours for training and is able to outperform the BLIP-2.
* Various qualitative examples are shown along with ablation studies to measure the effectiveness of the two-stage training procedure.
* Limitations of the MiniGPT-4 (in terms of hallucinations and spatial relations) are discussed.

**Weaknesses:**

* Benchmark is very limited. Rigorous benchmarking on different downstream tasks is needed. For reference, MMBench, TextVQA, etc.
* Only one baseline is reported. However, for a holistic understanding of the approach, more baselines are needed.

Some of the missing References:
* Liu, Haotian, Chunyuan Li, Qingyang Wu, and Yong Jae Lee. "Visual instruction tuning." arXiv preprint arXiv:2304.08485 (2023).
* Mou, Chong, Xintao Wang, Liangbin Xie, Jian Zhang, Zhongang Qi, Ying Shan, and Xiaohu Qie. "T2i-adapter: Learning adapters to dig out more controllable ability for text-to-image diffusion models." arXiv preprint arXiv:2302.08453 (2023).

**Questions:**

* Instead of having the data post-processing steps for stage 2 training, why not use GPT-4 itself to get the image captions and perform knowledge distillation as stage 2? Getting ~3000 such image-caption pairs should not be that costly.
* Does MiniGPT-4 apply a unique projection layer to all output vectors corresponding to the learned queries? Or some other processing steps are involved?

---

> ### Author Response · Authors · 2023-11-21
> **Response**
>
> Thank you for your valuable feedback! We response to the questions as follows:
>
> **Q1 Rigorous benchmarking on different downstream tasks is needed. For reference, MMBench. More baselines are needed.**
>
>
> As mentioned by Reviewer 7oDs, this project has been evaluated in the MMBench benchmark paper, and also in other benchmarks like MME and VisIT-Bench, together with similar baseline methods like LLaVa and InstructBLIP.  Kindly refer to Q0 in the general response for a detailed discussion.
>
>
> **Q2 Why not use GPT-4 itself to get the image captions and perform knowledge distillation as stage 2?**
>
> Kindly refer to Q3 in the general response for the new experiment.
>
>
> **Q3 Does MiniGPT-4 apply a unique projection layer to all output vectors corresponding to the learned queries? Or some other processing steps are involved?**
>
> The projection layer is applied to Q-former’s output vectors (tokens), which correspond to the Q-former’s internal learnable queries. This projection layer converts each of the output tokens individually to match the input size required by the LLM model.

---

> > ### Comment · Reviewer_TFFX · 2023-11-22
> > **Rebuttal Response**
> >
> > Thank you for the response and for performing new experiments (especially, gpt-4v). I recognize that gpt4-v might not have been available to everyone at this work time and I appreciate the quick experimental comparison.
> >
> > That said, the presentation still needs some improvements. I would appreciate it if the authors could include the above-mentioned benchmarking results in the main paper as this is a must-have at this stage of the submission.
> >
> > Most of my concerns are resolved. I'm increasing the score to 5 (would be happy to increase it to 6 if the presentation is improved).

---

> > > ### Author Response · Authors · 2023-11-22
> > >
> > > Thanks for your comment! We have updated the paper to include all new experiments for better presentation. New content appears in blue font.

---

### Official Review · Reviewer_7oDs · 2023-10-31

**Soundness:** 2 fair
**Presentation:** 3 good
**Contribution:** 3 good
**Rating:** 5
**Confidence:** 4

**Summary:**

This work introduces MiniGPT-4, which combines a frozen visual encoder with a frozen advanced LLM, Vicuna, through a single projection layer. Remarkably, MiniGPT-4 exhibits advanced multi-modal abilities similar to GPT-4, extending to generating detailed image descriptions, creating websites from hand-drawn drafts, writing stories and poems inspired by images, and providing cooking instructions based on food photos. However, initial training on short image captions led to issues like repetition and fragmentation in the generated text. To remedy this, the authors curated a detailed image description dataset for a second stage of fine-tuning, significantly enhancing the model’s reliability and overall performance. They have made their code available for further research and validation.

**Strengths:**

This submission is one of the earliest attempts at open-source reproduction of large vision-and-language models. It provides a parameter-efficient solution to this difficult problem, and makes use of the existing open-source models as much as possible, which inspires many follow-up works.

**Weaknesses:**

There are two main weaknesses with regard to the model structure and the evaluation setup. My evaluation of this submission will be greatly improved if the authors properly address these two main issues.


**(1) Model Structure:**
The introduction of MiniGPT-4 in the submission presents a novel approach in the domain of vision-and-language models. However, a noticeable limitation lies in its capacity to process only a single visual input per run. This design contrasts with other large-scale models such as Flamingo and GPT-4, which have demonstrated capabilities in handling multiple visual inputs simultaneously, as acknowledged in the related work section of the submission.

Additionally, the literature review appears to omit significant works [1,2,3,4] in the realm of large vision-and-language models that facilitate in-context learning with multiple visual inputs. The inclusion of these works could provide a more comprehensive backdrop for the MiniGPT-4, situating it within the broader context of ongoing research in the field.

The MiniGPT-4’s visual encoder is founded on the Q-Former architecture, converting each visual input into a fixed-length learned visual query. With reference to the BLIP-2 paper, this length is specified as 32 dimensions. Considering that this is considerably shorter than the maximum input length supported by the LLM backbone, there seems to be an opportunity to extend the model’s capacity to accommodate multiple visual inputs. An exploration into why the MiniGPT-4 is constrained to single image inputs, alongside potential avenues for extending its flexibility and application range, would strengthen the submission.

**(2) Evaluation Setup:**
The submission primarily focuses on showcasing qualitative results derived from the MiniGPT-4, complemented by various ablation studies. However, there is an apparent gap in benchmark comparisons with other established large vision-and-language models. A number of benchmarks [5,6,7], encompassing diverse aspects and task setups related to vision-and-language evaluation, have included results from MiniGPT-4 alongside comparisons with other models. The incorporation of such direct comparisons would offer a more definitive assessment of MiniGPT-4’s performance, enhancing the overall credibility of the evaluation presented.



**Missing references**:

[1] OpenFlamingo: An Open-Source Framework for Training Large Autoregressive Vision-Language Models

[2] Otter: A Multi-Modal Model with In-Context Instruction Tuning

[3] mPLUG-Owl: Modularization Empowers Large Language Models with Multimodality

[4] MIMIC-IT: Multi-Modal In-Context Instruction Tuning

[5] MME: A Comprehensive Evaluation Benchmark for Multimodal Large Language Models

[6] MMBench: Is Your Multi-modal Model an All-around Player?

[7] VisIT-Bench: A Benchmark for Vision-Language Instruction Following Inspired by Real-World Use

**Questions:**

(1) What is the maximum input and output length for the training and inference stage?

(2) In Section 3.2, the authors mention they would examine if the generated sentence exceeds 80 tokens or not – how is the length of 80 determined? Is it through empirical observations?

(3) What’s the length of the learned visual queries from the Q-Former in Mini-GPT4?

---

> ### Author Response · Authors · 2023-11-21
> **Response**
>
> Thank you for your valuable feedback! We response to the questions as follows:
>
> **Q1 Extend the model’s capacity to accommodate multiple visual inputs.**
>
> Kindly refer to Q1 in the general response section.
>
>
> **Q2 Evaluation Setup**
>
>
> Kindly refer to Q0 in the general response section.
>
>
>
> **Q3 What is the maximum input and output length for the training and inference stage?**
>
> The context length is limited by the LLM itself. For the vicuna version we use which is based on Llama 1, the context length is 2048 tokens (input + output). To ensure the training process does not exhaust GPU memory, we truncate the output to a maximum of 160 tokens if it exceeds this limit during training.
>
>
>
> **Q4 In Section 3.2, the authors mention they would examine if the generated sentence exceeds 80 tokens or not – how is the length of 80 determined? Is it through empirical observations?**
>
>
> The choice of this threshold is grounded in empirical observations. During our initial explorations, we manually examined the sentences generated and noticed that descriptions shorter than 80 tokens tended to be incomplete. Consequently, we established the threshold at 80 tokens.
>
>
>
> **Q5 What’s the length of the learned visual queries from the Q-Former in Mini-GPT4?**
>
> It is 32.
>
>
> **Q6 Missing reference.**
>
> We have updated the related work section in the paper (highlighted in blue) to include more related references.

---

### Author Response · Authors · 2023-11-21
**General Response**

We thank the reviewers for their insightful feedback. We are encouraged by the reviewers' recognition of our project's good contributions (Reviewer 7oDs, TFFX, A8a5), its status as one of the earliest attempts at an open-source large vision-language model (Reviewer 7oDS, TFFX), and its inspiration for many subsequent works (Reviewer 70DS). Additionally, reviewers appreciated our method's simplicity (Reviewer A8a5) and efficiency (Reviewer 70DS, TFFX), found our experiments interesting (Reviewer A8a5), noted the diverse capabilities of our model (Reviewer A8a5), and acknowledged that our project enhances the reproducibility and transparency of research (Reviewer auSX).
We address common questions raised by the reviewers here and respond to specific queries in each reviewer's section. All feedback will be incorporated accordingly.



**Q0 A number of benchmarks including MMBench and MME [1, 2] have included results from MiniGPT-4 alongside comparisons with other models. The incorporation of such direct comparisons would offer a more definitive assessment of MiniGPT-4’s performance.**

Reviewers TFFX and auSX have expressed interest in the model's performance on benchmarks like the referred MMBench and its comparison with other baselines such as Llava [3] or InstructBLIP [4]. As Reviewer 7oDs highlighted, benchmarks like MMBench [1] and MME [2] have incorporated MiniGPT4 in the comparison as one of the first works in this space and conducted evaluations alongside other models in their papers. Here, we do a simple analysis on their evaluations.

**In the MMBench benchmark**, which concentrates on visual reasoning, MiniGPT-4 demonstrates competitive performance compared to contemporary methods such as InstructBlip. Results from MMBenchmark are listed below. It surpasses InstructBlip in several key areas: logical reasoning (LR), fine-grained perception for single instance (FP-S), and fine-grained perception across instances (FP-C). Additionally, MiniGPT-4 achieves competitive results in relation reasoning (RR), attribute reasoning (AR), and coarse perception (CP).

| Model          | Overall |  LR  |  AR  |  RR  | FP-S | FP-C |  CP  |
|----------------|:-------:|:----:|:----:|:----:|:----:|:----:|:----:|
| OpenFlamingo   |  4.6%   | 6.7% | 8.0% | 0.0% | 6.7% | 2.8% | 2.0% |
| VisualGLM      |  38.1%  | 10.8%| 44.3%| **35.7%**| 43.8%| 23.4%| 47.3%|
| LLaVa          |  38.7%  | 16.7%| 48.3%| 30.4%| 45.5%| 32.4%| 40.6% |
| InstructBlip   |  **44%**   | 19.1%| **54.2%**| 34.8%| 47.8%| 24.8%| **56.4%**|
| MiniGPT-4      |  42.3%  | **20.8%**| 50.7%| 30.4%| **49.5%**| **26.2%**| 50.7%|

---

> ### Author Response · Authors · 2023-11-21
> **General Response Part II**
>
> **In the MME benchmark**, which focuses more on advanced multi-modal reasoning abilities like visual code reasoning and counting, MiniGPT-4 exhibits impressive performance. The MME benchmark is based on the Elo rating system. Here, we list the rank of MiniGPT-4 in tasks where MiniGPT-4 is top 4 and tasks where MiniGPT-4 is bottom 5.
>
> In this context, MiniGPT-4 has achieved the top rank in the overall leaderboard for cognition and is ranked first in 2 out of 14 subtasks (code reasoning, position reasoning). It also secured top 3 positions in 2 out of 14 subtasks (Counting, Color recognition), and top 4 in 2 out of 14 tasks (OCR, Numerical Calculation), among 11 other baselines including LLaVa, mPLUG-Owl, InstructBLIP, VisualGLM, Otter, etc.
>
> However, MiniGPT-4 does not perform as well in subtasks like Poster Recognition (rank 10), Artwork Recognition (rank 9), and Scene Recognition (rank 9), and Landmark Recognition (rank 8). These results suggest that MiniGPT-4 excels in cognition-related multi-modal tasks but is relatively weaker in fine-grained recognition tasks such as movie poster or artwork recognition. The strength of MiniGPT-4 in cognition tasks may be attributed to the frozen LLM component. Since the LLM isn't fine-tuned, its reasoning ability remains unaltered during multi-modal training.
> The comparatively weaker performance in fine-grained recognition tasks might also be a consequence of the frozen LLM. Without fine-tuning, aligning the visual module with the LLM for fine-grained and less frequently occurring concepts or objects, like artworks, can be challenging. Identifying ways to better balance cognition-related tasks with fine-grained recognition tasks presents a significant research opportunity, and we aim to explore this in future studies
>
>
> | Tasks                  | Blip-2 | InstructBlip | mPLUG-Owl | Llava | MiniGPT-4 |
> |-----------------------|:------:|:------------:|:---------:|:-----:|:--------:|
> | Cognition             |   3    |      2       |     4     |   9   |     **1**    |
> | Code                  |   2    |      4       |     4     |   6   |     **1**    |
> | Position              |   2    |      3       |     6     |   6   |     **1**    |
> | Counting          |   2    |      **1**       |     7     |   7   |     3    |
> | Color             |   2    |      **1**       |     7     |   7   |     3    |
> | OCR               |   **2**    |      7       |     8     |  11   |     4    |
> | Numerical Calculation |   7    |      7       |     **3**     |   5   |     4    |
> | Poster                |   **1**    |      3       |     2     |  11   |    10    |
> | Artwork               |   **1**    |      2       |     3     |  11   |     9    |
> | Landmark              |   3    |      6       |     **1**     |  10   |     8    |
> | Scene                 |   4    |      **1**      |     6     |  11   |     9    |
>
>
>
> [1] MMBench: Is Your Multi-modal Model an All-around Player?
>
> [2] MME: A Comprehensive Evaluation Benchmark for Multimodal Large Language Models
>
> [3] Visual Instruction Tuning, https://arxiv.org/abs/2304.08485
>
> [4] InstructBLIP: Towards general-purpose vision-language models with instruction tuning, https://arxiv.org/abs/2305.06500
>
>
>
> **Q1 Extends the model’s capacity to accommodate multiple visual inputs.**
>
> The MiniGPT-4 architecture design supports multiple image inputs. We tested this capability by fine-tuning MiniGPT-4 on VideoInstruct100K. For visual content representation, we chose 50 frames from each video and included any available subtitles. The model's final input comprised these frames and subtitles. We fine-tuned MiniGPT-4 using 1,200 video question-answer pairs, representing 1.2% of the original dataset's size. Its video understanding was assessed using the Video-based Generative Performance Benchmark, developed by Video ChatGPT. Contrary to our expectations, the results revealed that MiniGPT-4, fine-tuned for video, surpassed the state-of-the-art baseline, Video ChatGPT, in four of five metrics: correctness, detail, contextual understanding, and temporal understanding. It also performed well in consistency, despite being trained with only 1.2% of Video ChatGPT's data. These findings indicate that MiniGPT-4 is adept at processing multiple visual inputs.
>
> | Metric                    | Video Chat | Llama Adapter | Video LLama | Video-ChatGPT | MiniGPT-4 |
> |---------------------------|:----------:|:-------------:|:-----------:|:-------------:|:---------:|
> | Correctness of Information|    2.23    |      2.03     |     1.96    |      2.40     |    **2.68**   |
> | Detail Orientation        |    2.50    |      2.32     |     2.18    |      2.52     |    **2.76**   |
> | Contextual Understanding  |    2.53    |      2.30     |     2.16    |      2.62     |    **3.20**   |
> | Temporal Understanding    |    1.94    |      1.98     |     1.82    |      1.98     |    **2.26**   |
> | Consistency               |    2.24    |      2.15     |     1.79    |      **2.37**     |    2.18   |

---

> ### Author Response · Authors · 2023-11-21
> **General Response Part III**
>
> **Q2 In Table 2, the ChatGPT based metric doesn’t consider multiple candidate ground-truth captions.**
>
> Here, we introduce a new metric based on GPT4. We utilize GPT-4 turbo (gpt-4-1106-preview) to assess whether the generated descriptions capture the content of each ground truth caption individually. In the COCO dataset, each image is accompanied by 5 ground truth captions. For every image, we calculate the number of captions covered by the generated descriptions and then average this count across all 5000 test images to derive the final score. Our experimental results shown in the table below demonstrate that MiniGPT-4 covers more ground truth captions compared to Blip-2.
>
> | Metric                 | Blip2 | MiniGPT-4 |
> |------------------------|:-----:|:---------:|
> | # Covered GT Captions  | 1.96  |   2.22    |
>
>
>
>
> Here is the prompt we use in GPT-4 turbo
>
> ```
> Given a test image description and a list of gt image caption,
> verify whether the information in gt caption is included in the test description.
> The answer should be yes or no.
> Input is in this format:
> Test: (test sentence)
> 1: (gt1)
> 2: (gt2)
> 3: (gt3)
>
> you need to answer yes or no for each gt in the following format:
> 1: (yes/no)
> 2: (yes/no)
> 3: (yes/no)
> ```
>
>
> **Q3 Instead of having the data post-processing steps for stage 2 training, why not use GPT-4 itself to get the image captions and perform knowledge distillation as stage 2?**
>
> We agree with the reviewer that this suggestion is reasonable. GPT4-V was not available during the time when this project was developed, but we can now do this. We conducted new experiments using an image-text pair dataset collected by LAION, available at [https://huggingface.co/datasets/laion/gpt4v-dataset]. The image descriptions in this dataset were generated by GPT-4v. Currently, the dataset size is increasing, and at the time of our experiments, it contained 2,000 image-text pairs.
>
> We used this dataset to replace the original one in stage-2, while maintaining all other fine-tuning hyperparameters. The model was fine-tuned starting from the same stage 1 weights as the original version.
> We evaluated this variant using the metric we proposed in Q2. The experimental results below indicate that fine-tuning on the image descriptions generated by GPT-4v can enhance the model's performance.
>
>
> | Metric                 | MiniGPT-4 | MiniGPT-4(stage2-GPT4v) |
> |------------------------|:---------:|:----------------:|
> | # Covered GT Captions  |   2.22    |       2.26       |
>
>
> **Q4 Is it possible to use less data from the combined dataset to train the projection layer in the first pertaining stage? It is better to provide insights on the size data necessary to align the vision and text space.**
>
> We conducted a new ablation analysis focusing on the volume of training data used in the first stage. We selected checkpoints from midway through the first stage of training, corresponding to 10%, 30%, and 50% of the original stage 1 training duration. These checkpoints were then fine-tuned using the stage 2 training process.
>
> The experimental results demonstrate that reducing the first stage training data to 10% results in a significant performance decline. However, with 30% of the original first stage training, which is approximately 1.5 million image-text pairs and requires 3 hours of training on 4 A100 GPUs, we can achieve performance close to that of the original model. We don’t observe further improvement after reaching 50% of the first stage training. This finding suggests that the learable capacity (the one linear mapping layer), may reach saturation at this point.
>
>
> |        Metric         | 10% Stage1 | 30% Stage1 | 50% Stage1 | 100% Stage1 |
> |-----------------------|:----------:|:----------:|:----------:|:-----------:|
> | # Covered GT Captions |    1.62    |    2.15    |    2.26    |     2.22    |
>
>
> **Q5 Hallucination Comparison**
>
> Here, we extend table 5 in the main paper to include more baseline models in the hallucination evaluation. Compared to contemporary methods like Llava or mPlug-Owl, MiniGPT-4 generates longer descriptions with fewer hallucination.
>
> | Model  | $Chair_i$ | Avg. Length |
> |--------------------|:-------:|:-----------:|
> | Blip-2          |   1.3   |     6.5     |
> | mPLUG-Owl  |  30.2   |    98.5     |
> | LLaVa            |  18.8   |    90.7     |
> | MultiModal-GPT    |  18.2   |    45.7     |
> | MiniGPT-4 (short)  |   7.2   |    28.8     |
> | MiniGPT-4 (long)   |   9.6   |    175      |

---

### Meta-Review · Area_Chair_cV15 · 2023-12-16

**Metareview:**

**Scientific Claims and Findings**:

The paper introduces MiniGPT-4, a vision-language model combining a frozen visual encoder (with Q-former) with the large language model Vicuna through a single projection layer. MiniGPT-4 demonstrates advanced multi-modal abilities similar to GPT-4V, including image description, website creation from hand-drawn drafts, story and poem generation inspired by images, and cooking instructions based on food photos. The authors address issues arising from initial training on short image captions by curating a detailed image description dataset for fine-tuning, significantly enhancing model performance. The code is open-sourced for further research.

**Strengths of the Paper**:

- Innovative Approach: The paper presents a novel approach to combining vision and LLM, aligning a pre-trained vision encoder with a language model through a two-stage training process.
- Reproducibility: The provision of open-sourced model checkpoints and datasets enhances the reproducibility and transparency of the research, contributing to the scientific community.
- Advanced Abilities: MiniGPT-4 showcases advanced multi-modal capabilities in various tasks, providing qualitative evidence of its potential effectiveness in challenging vision-language tasks.

**Weaknesses of the Paper**:

- Evaluation Setup: The paper lacks comprehensive benchmark comparisons with more recent vision-language models. Comparisons with more recent models and larger benchmarks would strengthen the credibility of the results.
- Quantitative Evaluation Issues: The quantitative evaluation setup lacks convincing elements. The choice of baseline and evaluation metrics raises concerns about the completeness and fairness of the assessment. The paper could benefit from additional benchmarks and more rigorous quantitative comparisons.
- Misleading Claims: Some claims in the paper are considered misleading, such as the assertion that there are no equivalent datasets for multi-modal instruction tuning, and the implication that MiniGPT-4 achieves similar results with or without Qformer. These claims should be revised for accuracy.

**Justification For Why Not Higher Score:**

Limited evaluation on benchmarks.

**Justification For Why Not Lower Score:**

MiniGPT4 is the one of the first open-source projects to inspire the community to build large multimodal models. As a research prototype, it has successfully demonstrated an effective way to build GPT-4V like models, including data creation and model training.

---

### Decision · Program_Chairs · 2024-01-16

Accept (poster)